# From Cell Populations to Molecular Complexes: Multiplexed Multimodal Microscopy to Explore p53-53BP1 Molecular Interaction

**DOI:** 10.3390/ijms25094672

**Published:** 2024-04-25

**Authors:** Simone Pelicci, Laura Furia, Pier Giuseppe Pelicci, Mario Faretta

**Affiliations:** 1Department of Experimental Oncology, European Institute of Oncology IRCCS, 20139 Milan, Italy; simone.pelicci@ieo.it (S.P.); laura.furia@ieo.it (L.F.); piergiuseppe.pelicci@ieo.it (P.G.P.); 2Department of Oncology and Hemato-Oncology, University of Milan, 20122 Milan, Italy

**Keywords:** DNA damage response, 53BP1, p53, cell cycle, fluorescence microscopy, super-resolution microscopy, DNA PAINT microscopy, image analysis

## Abstract

Surpassing the diffraction barrier revolutionized modern fluorescence microscopy. However, intrinsic limitations in statistical sampling, the number of simultaneously analyzable channels, hardware requirements, and sample preparation procedures still represent an obstacle to its widespread diffusion in applicative biomedical research. Here, we present a novel pipeline based on automated multimodal microscopy and super-resolution techniques employing easily available materials and instruments and completed with open-source image-analysis software developed in our laboratory. The results show the potential impact of single-molecule localization microscopy (SMLM) on the study of biomolecules’ interactions and the localization of macromolecular complexes. As a demonstrative application, we explored the basis of p53-53BP1 interactions, showing the formation of a putative macromolecular complex between the two proteins and the basal transcription machinery in situ, thus providing visual proof of the direct role of 53BP1 in sustaining p53 transactivation function. Moreover, high-content SMLM provided evidence of the presence of a 53BP1 complex on the cell cytoskeleton and in the mitochondrial space, thus suggesting the existence of novel alternative 53BP1 functions to support p53 activity.

## 1. Introduction

From its birth, optical microscopy has been an instrumental tool for the study of the living world. After surpassing the diffraction barrier, with the advent of the first nanoscope, macromolecular complexes could be explored in situ in their natural environment [1,2]. Among super-resolution technologies, single-molecule localization microscopy (SMLM) with PALM, STORM, PAINT, and MINFLUX [3], up to their most powerful variants, i.e., RESI [4] and MINSTED [5], introduced a completely different approach to imaging. Unlike traditional fluorescence microscopy imaging, image formation is not simply achieved by the direct storage of emitting signals.

In SMLM, with appropriate probes and environmental conditions, only few isolated molecules stochastically undergo light emission at a certain time. The position of single emitters is instead calculated from the fitting of the corresponding Point Spread Function (PSF) with precision that depends on the number of collected photons. The process of localization is then iterated for thousands of frames to reveal the distribution of the fluorescent molecules in the field of view. The calculated coordinates for each molecule are then employed to generate a topographic map, reaching localization precision from tens to few nanometers in the case of the most powerful techniques. Historically, Photo-Activated Localization Microscopy (PALM) [6] paved the way for the development of the most modern SMLM technologies. However, PALM obtains high spatial resolution thanks to photoactivation processes of selected fluorescent molecules, often requiring the genetic manipulation of the sample for the expression of photoactivatable proteins. Even if new photoactivatable and photoconvertible molecules are available nowadays for conjugation to antibodies, the localization process based on progressive photobleaching or photoconversion still call for relatively long observation times, thus limiting the potential of the technique. The introduction of a chemically active environment able to temporally modulate the photophysical properties of emitters is the base of stochastic optical reconstruction microscopy (STORM) [7] that, with its derivative direct STORM (dSTORM) [8], contributed to the diffusion of SMLM technologies in applied biomedical research. The ability to perform an on–off transition from an emission prone to a dark state together with the use of selected buffers imposes important restrictions on the number of simultaneously analyzable channels, while the photobleaching effects makes the evaluation of the fraction of fluorescent molecules really localized at the end of the process particularly difficult. DNA Point Accumulation for Imaging in Nanoscale Topography (DNA PAINT) [9] removed some of these limitations by using the binding–unbinding transient immobilization of fluorescent DNA oligos to illuminate the biomolecules. DNA PAINT generally requires, with respect to STORM, longer exposure times to reach high localization precision but it has the enormous advantage of a limited influence of photobleaching and, as mentioned below, contributes to increasing the number of analyzable channels in a sample. MINFLUX [10] and its recent sister technology, MINSTED [5], adopt a reversed point of view in the localization of molecules. Instead of looking at the intensity peaks, these technologies search for emitters by beam scouting for a minimum number of emitted photons. Traditional SMLM obtains high-precision measurements by asking for high number of photons to the sample, in the MINFLUX framework, this requirement is satisfied by the light provided by the localizing doughnut-shaped laser beam, with a drastic cut in the duration of the measurement. However, the underlying technologies require complex and expensive hardware that strongly limits access to these microscopies in their infant phase for the application in the world of biomedical research.

Resolution Enhancement by Sequential Imaging (RESI) now provides a spatial resolution comparable to cryo-electron microscopy at the level of the nanometer by a novel DNA barcoding technique associated with multiple re-observations of the target structure with different tagging probes [4]. RESI, like the other traditional SMLM techniques, maintains the advantage of low-complexity instrumentation, making this approach available thanks to the limited costs.

Nonetheless, we have not assisted yet with the expected wide distribution of these novel technologies across research laboratories. One of the possible reasons for this resides in the price to pay when pushing for the highest spatial resolution. Working beyond the diffraction barrier implies a limitation in the other performances of the optical microscope such as sensitivity, speed, content (number of simultaneously observed parameters), and statistical sampling.

Moreover, the potential of the microscope has been frequently limited by man-driven usage. Human control results in diminished objectivity, low throughput, and semi- or poorly quantitative measurements. Consequently, efforts in moving towards an intelligent automated microscope, able to choose from the optimal observation conditions, are strongly desired and continuously growing. We recently published a novel pipeline to perform image-cytometry analysis by a standard motorized fluorescence microscope, reaching statistical sampling in the order of thousands of cells and content of several parameters [11]. Events can then be analyzed to quantitatively identify simple or complex target phenotypes, re-localized at the microscope stage and finally re-measured at the maximal resolution by 3D diffraction-limited imaging, confocal microscopy, or sub-diffraction analysis, by Structured Illumination or STORM super-resolution microscopy. At the same time, we proposed that correlative microscopy among different optical technologies can then be employed to remove the limitations in content typical of sub-diffraction microscopy, an approach that can be applied both to cells and tissues.

This pipeline was extended here by including single-molecule localization microscopy (SMLM) and sequential imaging. Many published papers successfully discuss the possibility of increasing the number of observed parameters by sequential immunostaining, reaching the visualization of up to ten channels with the resolution between 10 and 20 nanometers in the case of Exchange PAINT Microscopy [9]. However, the conjugation of antibodies to DNA oligos or fluorochromes can present issues, making the approach not easily applicable for all the laboratories. Using commercially available reagents, we demonstrated the possibility of obtaining a four-parameter analysis employing sequential measurements re-aligned thanks to the identification of fiducial points.

SMLM microscopy has been mainly employed to reveal details of biological structures in situ, localizing molecules with precision ten times smaller than the diffraction barrier. However, the difficulties in combining the analysis of multiple parameters in the same sample limited, until now, its use in the analysis of molecular interactions and their intracellular localization. The present work demonstrates the possibility of obtaining high-content super-resolution data to reveal the mutual compartmentalization of different molecular species according to the actual function they are exerting.

We applied the developed tools to further characterize the interaction between p53 and 53BP1 molecules. 53BP1 has been originally isolated in vitro as a p53 binder in two-hybrid screening [12]. After the discovery of its key role in the DNA Damage Recognition and Response (DDR) process [13,14,15], its function in supporting the action of the guardian of the genome has been frequently forgotten. However, many published papers shed light on the novel functions of 53BP1, linking in vivo 53BP1 protein to the p53-driven transcriptional program that determines cell fate [16,17,18,19]. In recent work, we applied automated image cytometry and correlative super-resolution microscopy to study the spatio-temporal dynamics of p53-53BP1 interaction following severe DNA damage by Ionizing Radiation [20]. Both spatial compartmentalization, with the formation of the putative p53-53BP1 complex out of the IR-induced γH2A.X foci, and temporal localization, with the detection of maximum interaction at late time points after irradiation when p53 was stabilized and damage foci reduced by the DNA-repair activity, suggested that 53BP1 can work to sustain the p53-controlled transcriptional program.

We thus focused on the molecular mechanisms at the base of the 53BP1 action in supporting the p53 transactivation function. Cells were classified according to (i) p53 and 53BP1 content, (ii) transcriptional activity by RNA Polymerase II phosphorylated on Serine 5 (Pol II-S5p), and (iii) DNA integrity by γH2A.X foci detection by diffraction-limited widefield microscopy and re-localized for super-resolution analysis. A two-step SMLM analysis was employed to monitor Pol II-S5p-p53, 53BP1-p53, Pol II-S5p-53BP1, and putative Pol II-S5p-53BP1-p53 interactions in and out of IR-induced foci, sustaining a model that includes 53BP1 in a putative complex with p53 as a transcription factor and basal transcription machinery. We also demonstrated that p53-53BP1 spatial compartmentalization can lead to the formation of putative complexes in the cytoplasm, thus suggesting the hypothesis that 53BP1 can also exert additional functions linked to its shuttling nature.

## 2. Results

### 2.1. Multivariate Quantitative Analysis of p53, 53BP1, and Transcriptional Activity

Much evidence has shown that 53BP1 sustains the p53 activity in the cell fate decision [16,17,18,19,20] as a complementary function to its well-known role in DNA Damage Recognition and Response (DDR). To investigate if 53BP1 directly participates in p53 transactivation activity, we first employed an image-cytometry pipeline to quantitatively analyze the relationships between p53 content, IR-induced foci and DDR kinetics, and transcriptional activity measured by the amount of RNA Polymerase II phosphorylated on Serine 5 (Pol II-S5p), a marker of transcriptional initiation.

After 24 h from irradiation, most of the cells were arrested in the G2 phase of the cell cycle, with a minor fraction in G1 and an almost negligible number of cells undergoing DNA replication, as evidenced by the DNA content distribution and EdU incorporation (Figure 1a,b).

The simultaneous measurement of the content of KI67 protein showed that diploid cells entered the G0 phase of the cell cycle, significantly downregulating the expression of the proliferation marker (Figure 1c). Arrested G2 cells maintained levels comparable to the ones exhibited by the exponentially growing counterpart. This preserved attitude to proliferation was probably the reason for the appearance of a tetraploid population generated by aberrant mitosis that led to micronuclei formation 48 h after irradiation (Appendix A).

The appearance of a quiescent fraction of the population was also underlined by a significant reduction in the transcriptional activity (Figure 1d), while, as previously observed for the KI67 levels, the Pol II-S5p content was only slightly modified in the G2 phase.

The recognition of the induced damage led to p53 stabilization at 24 h after the irradiation, both in quiescent and in proliferating cells (Figure 1e).

Finally, the 53BP1 content was upregulated independently of its localization. Both the amount of protein in foci and diffused in the nucleoplasm (Figure 1h,f, respectively) was significantly higher than the basal level.

The intranuclear distribution of DD foci, measured by 53BP1 protein content and localization, showed the heterogenous progression in the DDR. As previously described [20], irradiation caused the formation of a high number of foci accumulating γH2A.X and 53BP1 (Figure 1g,h). As DNA repair progressed, a fraction of cells showed IR foci of increased size, in agreement with the model of DNA repair factories formed by clustering damaged DNA (Figure 1i,j and Appendix A). Accordingly, in the late phases of DDR, the average number of IR spots per nucleus was simultaneously reduced, while the integrated intensity over all the foci present in the nucleus was maintained independently of the number of detected spots. The phenotype was mainly manifested in the diploid DNA content population that stopped its proliferation activity, but it was also observed in a fraction of the arrested G2M phase.

Thus, automated fluorescence microscopy for image cytometry provided a quantitative description of all the events following the induction of the DD: (i) arrest of the cell cycle (G2 block) with loss (formation of a diploid G0 fraction) of the proliferation ability, (ii) recognition and repair by formation and clustering of the injured genome in foci accumulating H2A.X and 53BP1, and (iii) stabilization of p53 (iv) modulation of the transcriptional activity. Arrested cells maintained transcriptional levels comparable to the ones of the control except for the diploid subpopulation that exhibited reduced levels with respect to their proliferating counterpart.

### 2.2. Multivariate Quantitative Analysis of p53-53BP1 Interaction and Transcriptional Activity

To investigate if the p53-controlled transcriptional program can have a quantitative, clearly identifiable impact on the general transcriptional activity following stress, we performed a comparative analysis of the three players. Pol II-S5p, p53, and 53BP1 did not show any striking correlation (Figure 2a–c) in their quantitative distribution per cell. So, transcriptional activation by p53 cannot be deconvolved, as it is masked in the general RNA PolII activity.

However, the isolation of the population showing the highest amount of each of the three analyzed molecular species revealed a trend in the simultaneous accumulation of all the other components: cells with increased p53 content showed a trend in increasing their transcriptional activity and amount of nucleoplasmic 53BP1 (Figure 2a, third dot plot, blue spots), a higher amount of phosphorylated PolII associated with increased p53 and 53BP1 content, and similarly highly positive 53BP1 cells (Figure 2b,c, third dot plot, blue spots).

To validate the observed correlative trend, obtaining a more precise quantification of the putative link between DNA transcription and 53BP1-p53 interaction, we included in our automated cytometry analysis a proximity ligation assay (PLA) [21] to monitor the formation of the putative 53BP1-p53 complex (Figure 2d–f).

A relevant number of PLA foci, indicating a proximity of the two targets less than 50 nm, has been detected in all the cells with a distribution proportional to the amount of p53.

The 53BP1 function follows a spatial–temporal compartmentalization: the p53-53BP1 interaction mainly localized outside Ionizing Radiation (IR)-induced foci [20] where the canonical DDR activity of 53BP1 took place. Accordingly, the number of PLA spots located in proximity of IR foci, detected by 53BP1 accumulation, was dramatically reduced (Figure 2d,e). In agreement with the previously observed protein-content distribution, monitoring of the transcriptional activity in cells with an increased number of interaction spots correlated with higher levels of RNA Pol II-S5p (Figure 2f, blue dots).

As further proof of the existence of the link, we focused on p53 di-methylated on lysine 370. This post-translational modification stabilizes the binding to 53BP1 that, in this context, may function as coactivator of the guardian of the genome [22]. Even if the detected number of PLA dots was lower, the distribution of interacting spots was superimposable to the one measured for the total amount of p53, including both proximity to IR foci and influence on the transcriptional activity (Figure 2g–i).

### 2.3. Correlative PLA–Single-Molecule Localization Microscopy Analyis of p53-53BP1-Pol II-S5p

Even if image-cytometry analysis provided clues of its possible cooperative action, the only way to confirm a direct role of the p53-53BP1 putative complex on the transcriptional process is to perform a spatial analysis with a resolution able to reach the macromolecular aggregates range. PLA provided a first tool to investigate the tens of nanometer scale, demonstrating how the 53BP1-p53 interaction mainly took place far from the damaged genomic sites. However, the extension of the technique to three or more potential interactors is not so easily applicable [23].

Single-molecule localization microscopy (SMLM) can reach a localization precision less than 20 nm. However, SMLM is usually limited in the number of simultaneously detectable parameters.

We thus decided to couple the PLA assay to SMLM by a correlative microscopy pipeline previously developed in our laboratory [11]. DNA PAINT microscopy was employed for a two-channel analysis of RNA Pol II-S5p and γH2A.X to monitor, at high resolution, the transcriptional and DDR processes. Confocal microscopy was simultaneously used to detect PLA foci and the spatial condensation of chromatin. The number of molecular events inside and in proximity to the p53-53BP1 interaction sites was thus calculated. At the same time, the ultrastructure of damaged DNA regions was revealed and their relative position to the transcriptional events measured.

A quantitative diffraction-limited widefield analysis of the cell-cycle distribution of PLA foci was first calculated to allow the selection of the cell targets with the highest number of spots, similarly to what is shown in Figure 2. The re-localized cells were then processed for Confocal-DNA PAINT Microscopy correlative analysis (Figure 3) at maximum resolution.

PLA spots were distributed across highly decondensed chromatin and largely excluded from the remaining heterochromatic regions, i.e., the peri-nucleolar area and nuclear lamina (Figure 3a–c). γH2A.X localization inside IR foci revealed the chromatin distribution in damaged genomic regions, showing that the few detected transcription events were mainly located at the periphery and/or in empty spaces when imaged at single-molecule resolution (Figure 3b, insets 3–5 and relative line profiles).

Even if IR foci and PLA spots presented comparable sizes, in the range of 200–500 nm (Figure 3a,b), the relevant number of Pol II-S5p events detectable in the PLA regions (Figure 3b,c and Figure 4a,b) definitively demonstrated an association with the transcriptional process.

As previously performed for the image-cytometry diffraction-limited analysis, we repeated the PLA assay by looking at the di-methylated p53 on Lysine 370. The obtained results (Figure 3d–f and Figure 4c,d) confirmed the localization of the putative p53K370me2-53BP1 complex in decondensed chromatin regions and its association with the general transcriptional machinery.

The adopted image-cytometry correlative-microscopy pipeline thus allowed the following: (i) the selection of cell targets enriched in p53-53BP1 putative complexes; (ii) the description of the chromatin status by high-resolution confocal analysis, revealing how global decondensation was a requisite for both DNA repair and transcription; (iii) the ultrastructural description of IR foci and positioning of the transcription complexes inside and close to them; (iv) the relative localization of PLA and γH2A.X, confirming the two complementary functions of 53BP1 in DDR and p53 interaction; (v) the demonstration of a spatial proximity of p53, 53BP1, and RNA Polymerase II in a range less than 50 nm in the nucleoplasm.

### 2.4. Exchange DNA PAINT Microscopy to Study DDR and Transcriptional Activity

Even if correlative PLA versus DNA PAINT analysis could partially bypass the limitation in the content of SMLM, it showed only partially informative data on the putative complexes due to the diffraction-limited imaging of PLA. Moreover, PLA is also characterized by 30% limited efficiency and difficulties in determining the real number of detected macromolecular complexes [24].

High localization precision for all the involved targets, i.e., γH2A.X for DDR, Pol II-S5p as a transcription marker, p53, and 53BP1, was thus desirable. Jungmann and coworkers developed high-content sequential immunostaining applicable to DNA PAINT microscopy called Exchange PAINT [9]. We thus inserted a four-parameter Exchange PAINT protocol in our image-cytometry multimodal-microscopy pipeline to identify target cells and then to localize molecular complexes at the sub-diffraction level by a two-step, two-color sequential-detection experiment.

One difficulty related to DNA PAINT microscopy originates from its blinking imaging, ruled by the binding–unbinding process of the fluorescent DNA oligos. The selection of quantitative phenotypes by image-cytometry requires high statistical sampling and preliminary knowledge of the expression and distribution pattern of the molecular targets that cannot be consequently accomplished with PAINT staining. Taking advantage of the extremely high expression levels reached by 53BP1 after irradiation and of the optimal affinity of the employed reagents, we modified the staining procedure. Saturating conditions by DNA-conjugated secondary antibodies were not fully reached, thus allowing for the simultaneous detection of a fraction of antigens by standard fluorescent immune molecules. This way, it was possible to perform a preliminary identification of the targets: according to the previously performed image-cytometry studies (Figure 1 and Figure 2), we decided to focus on 4N DNA content cells with large foci to ensure higher transcriptional activity and a higher number of p53-53BP1 putative complexes (Figure 5a).

The previously isolated phenotype, characterized by foci aggregation and their reduction in number, was compatible with diminished DDR-related 53BP1 activity, having accumulated, at the same time, a relevant 53BP1 amount in the nucleoplasm. According to what has been observed until now, these conditions should maximize, together with the high p53 levels and transcriptional levels, the probability of visualizing 53BP1 activity in support of the p53 transactivation program. Diploid cells were not considered due to their entrance into the quiescent state with reduced transcriptional activity.

Four-color nuclear imaging allowed for the simultaneous reconstruction of the IR foci, obtaining information about the relationship between DNA damage and the transcription of injured DNA in foci. Ultrastructural analysis (Figure 5b,c) revealed how γH2A.X molecules were shielded by 53BP1, separating them from transcription factories and p53 molecules, in agreement with the fluid-phase separation model hypothesized for 53BP1 [25,26,27,28]. p53 was seldom observed inside an IR focus, despite recent work showing its involvement in DDR [29], but this event could be heavily influenced by the long time that passed after irradiation. However, even in this case, it showed proximity to the 53BP1 layer. The same considerations could be applied to the molecular distribution of Pol II-S5p. Consequently, tri-complexes, 53BP1, p53, and Pol II-S5p, have always been observed in low frequency when localized at the periphery of the damaged area. As expected, a much higher number of putative complexes have been instead identified in the nucleoplasm (Figure 5d), far from IR foci.

Analogous considerations could be extended to the single-molecule resolved distribution of the di-methylated form of p53 (Figure 5e–h): tri-complexes were always observed in the nucleoplasm, reinforcing the model of 53BP1-p53 interaction that supports transcriptional activity complementing the 53BP1 action in DDR.

### 2.5. Exchange DNA PAINT Microscopy Identify p53-53BP1 Interactions throughout the Cell

The stress response leads to p53 localization in different cellular compartments across the cell. Its action in regulating mitochondria metabolism and cell death induction has been extensively reported in the literature [30,31,32].

Previous works identified a DynLL1/LC8 interaction domain in 53BP1, making it a strong candidate as cargo-adapter protein for p53 transport across the cell [33]. Moreover, 53BP1 has also been associated with the regulation of mitochondrial homeostasis and clearance [34].

We thus decided to employ the Exchange DNA PAINT protocol to evaluate if the p53-53BP1 interaction could be observed outside the nuclear compartment, focusing on the cell cytoskeleton and mitochondrial network (Figure 6).

β-Tubulin distribution revealed strong 53BP1 accumulation on the filaments. 53BP1-accumulating regions present on the cytoskeleton were frequently accompanied by p53 enrichment, as evidenced by tubulin-signal segmentation (Figure 6b).

TOMM20 staining of the mitochondrial outer membrane revealed the expected p53 re-localization. Also in this case, compartment segmentation showed the accumulation of both p53 and 53BP1 protein with the formation of putative complexes characterized by events at distances closer than 20 nm (Figure 6c).

The extension of the protocol for high-content SMLM analysis thus provided supporting evidence for 53BP1-mediated p53 motility.

## 3. Discussion

Super-resolution microscopy revolutionized the world of optics by surpassing the diffraction barrier. This achievement was the result of considering the photophysical properties of the fluorescent molecules as active parts of the process, allowing the development of STED, PALM, STORM, and similar techniques that now provide a spatial resolution able to approach the performance of the electron microscope. However, different obstacles made it difficult for these novel technologies to spread into biomedical research laboratories. In some cases, the complexity of the required hardware and the consequent high costs strongly limit their availability for many potential interested users. Luckily, technological developments are rapidly contributing to the removal of this limiting aspect. However, some super-resolution techniques were built by relatively simple and accessible modifications of the fluorescence microscope.

Single-molecule localization microscopy (SMLM) reaches a resolution of tens of nanometers by just employing high-power laser sources and, eventually, specialized optics typical of Total Internal Reflection Microscopy. The large number of images required for the collection of a single sample unfortunately represents a big obstacle for “routine” use in applicative research. However, technology researchers are now paving the way for highly automated systems able to reach a throughput of thousands of analyzed cells [35,36].

On the other hand, classical SMLM approaches, i.e., PALM and STORM, suffer from a low number of simultaneously detectable fluorescence parameters due to their requirements regarding employable fluorochromes. DNA PAINT removed these limitations by sequential detection in Exchange PAINT microscopy [9].

Even if these new tools are easily retrievable nowadays, SMLM, in most cases, has been mainly employed for single-parameter analysis to investigate in situ cellular structures at high resolution. Relatively few attempts have been made to investigate biomolecular interactions even after the reached increase in the achievable content, i.e., the number of measurable parameters on a single sample.

Correlative microscopy allows a dynamic choice of the real spatial resolution to be achieved in every step of an experiment flow (e.g., statistical sampling, duration of the experiment, content) [11,37,38]. This way, all the performances of the microscope for the detection process can be optimized.

Here, we presented a pipeline to show the enormous potential of automated multimodal, multi-resolution correlative microscopy when working at a resolution in the order of tens of nanometers. Automated analysis-driven acquisition allows for the selection of quantitative and qualitative phenotypes producing both statistical reliability and high resolution. Adding an adapted Exchange PAINT protocol to this pipeline, we demonstrated how SMLM can be employed to (i) visualize the formation of putative macromolecular complexes and (ii) localize with precision in the order of 10–20 nanometers single molecules and complexes on intracellular structures.

The use of fluorescent nanodiamonds (FNDs) as sub-diffraction reference markers [39] allowed us to modify the Exchange DNA PAINT protocol originally developed by Jungmann and coworkers to make the procedure accessible even in the absence of a microfluidic system on the microscope, simplifying the hardware requirements. Exchange PAINT is based on a series of sequential imaging phases: a set of DNA oligos conjugated to standard fluorochromes binds (and unbinds) to the corresponding target sequences linked to the employed primary antibodies, thus generating the single-molecule events. The first group of fluorescent oligos is then washed away by microfluidics and replaced by the injection of new fluorescent DNA sequences. The process is then iterated until all the required parameters are covered. In our protocol, the washing and reporter–oligo replacement were performed at the bench. Efficient repositioning was possible thanks to the use of an ad hoc designed sample holder and the employment of FND as reference markers (Appendix A).

We also propose an alternative approach for obtaining information at the spatial scale of tens of nanometers based on the combined use of proximity ligation analysis, diffraction-limited confocal microscopy, and SMLM. PLA visualizes putative macromolecular complexes measuring the proximity of the target molecules in the range of about 50 nm, a value close to the resolution reached by super-resolution microscopy techniques. Even if PLA efficiency is limited, its employment allows for (i) the obtainment of relevant statistical sampling thanks to the count of the number of interaction spots in a population composed by thousands of events, (ii) the targeting of the cells enriched by putative complexes, and (iii) their analysis for the interaction with additional candidates by correlative confocal-DNA PAINT and/or STORM microscopy. PLA may also function as a guide for post-acquisition data analysis by providing a reference region for the counting and clustering of events. Finally, PLA spots by themselves provide a useful marker for both drift correction, allowing for the maximization of the localization precision and the spatial registration of the images from different microscopy modalities, e.g., confocal versus SMLM, or from different SMLM imaging acquisitions as for Exchange DNA PAINT.

We validated the developed tools by analyzing the putative interaction between p53 and 53BP1 protein. As already mentioned, 53BP1 is simultaneously a major effector of DDR and a key interactor of the guardian of the genome to sustain its transcriptional program for cell-fate decisions. The employment of the new multimodal multiresolution and high-content SMLM pipeline allowed the detection and localization of p53-53BP1 molecular complexes with precision of less than 20 nm. Even if the literature already reported evidence of this interaction, this is the first time, to our knowledge, that the putative p53-53BP1 complex has been visualized in cells. The detection of 53BP1-p53-PolIIS5p putative complexes confirmed that 53BP1 transcriptionally related function occurs far from damaged DNA sites. High-content Exchange PAINT also provided clues on the existence of compartmentalization-regulated 53BP1-p53 interactions. 53BP1-p53 putative complexes have been detected in the cytoskeleton, providing an in situ confirmation of the potential role of 53BP1 in the regulation of p53 shuttling through the cell. We also found 53BP1, in association with p53 or alone, in the mitochondria. 53BP1’s function in the control of mitochondria homeostasis has been previously hypothesized [34] based on the effects produced by the knock-out of the gene and without proving the presence of the protein in the compartment. This is the first proof, to our knowledge, of 53BP1’s presence in the mitochondria. These observations support a 53BP1-putative function as cargo for p53 and simultaneously suggest the existence of novel activities not necessarily related to p53.

Even if automated cytometry associated with multimodal correlative microscopy provides advantages in the reachable throughput of SMLM techniques, new developments are required to remove the limitations in their statistical sampling.

With this in mind, we are aware that the biological observations reported here may require further validation, considering their limited, even if increased, throughput, i.e., in the order of tens of cells. However, they can be seen as additional confirmation and refinements of already existing models. At the same time, we would like to underline that the main goal of the present work is to show the potential of the developed pipeline to favor the spreading of the SMLM technologies as a complement to the pure molecular biology approach.

Automated image cytometry coupled with multimodal high- and super-resolution microscopy can contribute to a new generation of fluorescence microscopy applications. The dynamic adaptation of the employed observation technique with the optimal spatial resolution and acquisition performances can have a profound impact on the management of experimental data. The automated selection of targets for high-resolution analysis in fixed samples enormously reduces storage efforts and makes the observation of rare phenotypes with increased statistical sampling possible [40]. For example, when coupled to artificial intelligence, the approach can have a big impact in the contribution of microscopy to pathology [41,42,43,44]. In basic biomedical research, the presented methodologies are now gaining an increasing space with event-driven acquisitions [40,45,46,47]. The simultaneous development of new computational tools is now providing solutions to dramatically power up the statistical sampling in SMLM [35,36,48], and the possibility of retrieving high-resolution sub-diffraction data from few acquired frames or even from a single snapshot [49] can offer new solutions to measure molecular dynamics in real time. At the same time, the introduction of novel microscopy techniques in the framework of acquisition-driven image cytometry can generate a great opportunity to complement and bypass present limitations. Super-resolution microscopy and SMLM are now indeed able to reconstruct the distribution of fluorescent molecules with a sub-nanometer resolution. However, it is necessary to keep in mind that what we are gaining is a view of tagging fluorescent molecules in a sample; label-free methods are thus strongly required to gain a more precise understanding of the molecular mechanisms that rule life. Fluorescence Lifetime Microscopy (FLIM) provides one of the examples of how microscopy can evolve in this direction. The temporal resolution achieved, together with its increased optical penetration when coupled to multi-photon excitation, provides a tool able to monitor events in vivo at the molecular level without any exogenous fluorescence carrier that can easily fit with optimal performance in the general context of event-driven intravital microscopy [50,51,52,53,54].

## 4. Materials and Methods

### 4.1. Cell Culture

MCF10A cells from the American Tissue Culture Collection (ATCC) were cultured in 50% DMEM High Glucose with stable L-glutamin (DMEM) (Euroclone) + 50% Ham’s F12 Medium (ThermoFisher Scientific, Waltham, MA, USA) containing 5% horse serum, 50 ng/mL penicillin/streptomycin (both from Euroclone, Milan, Italy), cholera toxin (Merck Life Science, Milan, Italy.), 10 mg/mL insulin (Merck Life Science, Milan, Italy), 500 ng/mL hydrocortisone (Merck Life Science, Milan, Italy), and 20 ng/mL EGF (Pepro Tech, Cranbury, NJ, USA,) at 37°C in 5% CO_2_. Cells were cultured on glass-bottom dishes (MatTek, Ashland, MA, USA). Growing cells were fixed for 10 min in 4% paraformaldehyde (wt/vol) to guarantee the exponential phase, whereas irradiated cells were exposed to 5-Gy radiation by an X-ray machine and fixed after 24 and 48 h.

### 4.2. Immunofluorescence of MCF10A Cells

Fixed MCF10A cells were washed and permeabilized for 10 min in a buffer containing 0.1% Triton X-100 (vol/vol) in PBS. EdU incorporation into DNA was detected following the manufacturer’s instructions by Biotin Azide (PEG4 carboxamide-6-Azidohexanyl Biotin) (ThermoFisher Scientific, Waltham, MA, USA) and Goat anti-biotin DyLight™ 800 (600145098, Rockland Immunochemicals, Pottstown, PA, USA). All the steps of the Click-iT reaction were performed at RT. After EdU detection, samples were incubated for 30 min in a blocking solution, 5% BSA (wt/vol) in PBS, and then overnight at 4 °C with primary antibodies in blocking solution. After washing in PBS, samples were incubated with secondary antibodies for 45 min. The following primary and secondary antibodies were employed: anti-53BP1 (ab36823, Abcam, Cambridge, UK) detected by Goat anti-Rabbit IgG (H+L) Cross-Adsorbed Secondary Antibody, Pacific Orange™ (ThermoFisher Scientific, Waltham, MA, USA), mouse anti-p53 Igg2a (sc-126, Santa Cruz Biotechnologies, Dallas, TX, USA) detected by Alexa Fluor® 488 AffiniPure™ Goat Anti-Mouse IgG, Fcγ subclass 2a specific (115-545-206, Jackson-immunoresearch, West Grove, PA, USA), rat anti-RNA Pol II Ser-5P (3e8-1 ChromoTek, Planegg, Germany) detected by Donkey anti-Rat IgG (H+L) Highly Cross-Adsorbed Secondary Antibody, Alexa Fluor™ Plus 555 (ThermoFisher Scientific, Waltham, MA, USA), mouse anti-phosphoH2A.X (ser39) (γH2A.X) Igg1 (613402, Biolegend, San Diego, CA, USA) detected by Alexa Fluor^®^ 647 AffiniPure™ Goat Anti-Mouse IgG, Fcγ subclass 1 specific (115-605-205, Jackson-immunoresearch, West Grove, PA, USA), Horizon™ V450 mouse anti-KI67 (561281, BD Biosciences, Franklin Lakes, NJ, USA). DNA was counterstained with Hoechst 33342, Trihydrochloride, and Trihydrate with a 10 mg/mL final concentration (ThermoFisher Scientific, Waltham, MA, USA). Finally, coverslips were then mounted in Slowfade Gold Antifade Mountant (ThermoFisher Scientific, Waltham, MA, USA).

### 4.3. In Situ Proximity Ligation Analysis (PLA)

Fixed MCF10A cells were washed and permeabilized for 10 min in a buffer containing 0.1% Triton X-100 (vol/vol) in PBS. Samples were then processed for an in situ proximity ligation assay (PLA) using NaveniFlex MR green detection reagent (Navinci, Uppsala, Sweden) according to the manufacturer’s instructions. The primary antibodies employed for PLA were rabbit anti-53BP1 (ab36823, Abcam, Cambridge, UK), mouse anti-p53 Igg2a (sc-126, Santa Cruz Biotechnologies, Dallas, TX, USA), rabbit Anti-Di-Methyl-TP53-Lys370 (STJ90115, St John’s Laboratory, London, UK), and mouse anti-53BP1 (612522, BD Biosciences, Franklin Lakes, NJ, USA). After the PLA reaction, the cells were processed for standard immunofluorescence for the detection of additional markers.

### 4.4. DNA PAINT Immunostaining

Sample were imaged by a commercial inverted Nikon Eclipse Ti2 microscope (Nikon instruments, Tokyo, Japan) equipped with an A1R confocal scan-head and N-SIM and N-STORM modules (Nikon instruments, Tokyo, Japan) and controlled by NIS Elements software (version 5.42.01). For widefield microscopy, an LED light source (pE-4000, CoolLED, Andover, United Kingdom) with 16 selectable wavelengths for widefield microscopy was employed. The light source for confocal imaging was a laser unit (LU-NV, Nikon instruments, Tokyo, Japan) equipped with 5 laser lines (405 nm (23.1 mW), 440 nm (25.5 mW) 488 nm (79.1 mW), 561 nm (79 mW), 647 nm (137 mW)), while a laser bench (L4Cc combiner, Oxxius S.A., Lannion, France) equipped with four high-power sources (405nm (216 mW), 488 nm (240 mW), 561 nm (240 mW), 640 nm (360 mW)) was employed for single-molecule localization experiments. A filter wheel (Optospin, Cairn Research Ltd, Faversham, Kent, UK) was placed in front of a CMOS camera (Dual ORCA Flash 4.0 Digital CMOS camera C13440, Hamamatsu, Tokyo, Japan) to acquire 16-bit scaled images (Widefield/DNA PAINT).

### 4.5. Microscope Setup

All data were acquired with a commercial inverted Nikon Eclipse Ti2 microscope (Nikon instruments, Tokyo, Japan), equipped with an A1R confocal scanhead and N-SIM and N-STORM modules (Nikon instruments, Tokyo, Japan). The fully motorized automated microscope was controlled by the NIS Elements software (version 5.42.01). The system performed multicolor widefield, confocal, and single-molecule localization imaging thanks to (i) an LED light source (pE-4000, CoolLED, Andover, UK) with 16 selectable wavelengths for widefield microscopy; (ii) a laser unit (LU-NV, Nikon instruments, Tokyo, Japan) equipped with 5 laser lines (405 nm (23.1 mW), 440 nm (25.5 mW) 488 nm (79.1 mW), 561 nm (79 mW), 647 nm (137 mW)) for confocal microscopy, and (iii) a laser bench (L4Cc combiner, Oxxius S.A., Lannion, France) equipped with four high-power sources (405 nm (216 mW), 488 nm (240 mW), 561 nm (240 mW), 640 nm (360 mW)) and two acousto-optic modulators for single-molecule microscopy). Emitted light was filtered by a filter wheel (Optospin, Cairn Research Ltd., Faversham, Kent, UK) and then was collected by a CMOS camera (Dual ORCA Flash 4.0 Digital CMOS camera C13440, Hamamatsu, Tokyo, Japan) set on a 16-bit scale detection modality (Widefield/DNA PAINT).

#### 4.5.1. Widefield Microscopy

A 60x Plan Apo 1.4 NA objective was employed in widefield microscopy, obtaining diffraction-limited imaging in the order of the collected wavelengths. The exposure time per each fluorescence channel maximized the dynamic range, avoiding saturation. Optimal values were set in a preliminary observation of randomly chosen positions (between 30 and 200 ms per frame). 

#### 4.5.2. Confocal Microscopy

A Nikon A1R confocal microscope with a 100x 1.49 NA Apochromat objective (Nikon instruments, Tokyo, Japan) was employed to obtain confocal images. The employed conditions minimized crosstalk in favor of an optimal signal-to-noise ratio. During correlative confocal-DNA PAINT, channel acquisitions were sequentially performed with the high-speed galvanometric scanning mirrors (4x Line Average) to maximize image quality in a reduced time. The pinhole aperture was set to 0.6 Airy Unit. Data were collected with a digital size of 1024 × 1024 pixels, with a pixel of ∼0.065 μm. Scanning pixel dwell time (7 fps) and laser power were set to limit the photobleaching effect. 

#### 4.5.3. DNA PAINT Microscopy

Single-molecule imaging was performed using the system described in previous works [11] using a Nikon CFI SR Apochromat TIRF 100× oil objective (1.49 NA). ATTO655 and Cy3b imagers were excited by using 561nm and 640nm laser wavelengths (L4Cc combiner, Oxxius S.A., Lannion, France). A multi-band dichroic mirror (C-NSTORM QUAD 405/488/561/647 FILTER SET; Chroma Technology Corporation, Bellows Falls, VT, USA), combined with 561 nm and 647 nm emission filters (Semrock Brightline^®^, IDEX Health & Science, West Henrietta, NY, USA), was used to filter the fluorescence excitation. The acquisitions of 2 channels (correlative experiments) and 4 channels (Exchange PAINT) were performed sequentially. Fluorescent nanodiamonds (40 nm size) conjugated to streptavidin (Adamas Nanotechnologies, Raleigh, NC, USA), previously incubated with cells as described below, were excited by the 488 nm laser line and collected every 1000 frames to correct the drift between frames and the shift among channels.

Z drift was compensated in real time by a hardware autofocusing system based on the reflection of a near-infrared light (Perfect Focus System (PFS), Nikon instruments, Tokyo, Japan). 

The number of frames and exposure time per channel depends on the density pattern of the immunostaining and imager concentration (at least 20,000 frames/channel with 90 ms of exposure time). Optimal imager concentrations were empirically determined according to the labeling density. The laser power was set according to the binding–unbinding rates of the imagers used. Additionally, the optimal imager concentration (and thus, the sufficiently low-fluorescence off-time) was necessary to ensure the proper number of binding events and thereby the robust detection of every docking strand during image acquisition. We employed 2–3 mL of imager buffer to maintain a constant concentration of imagers to minimize the negative effects of evaporation and photobleaching.

#### 4.5.4. Exchange PAINT 

For fluid exchange, a home-made clamping holder was constructed and mounted to the microscope stage, as shown in Appendix A. A schematic representation of the Exchange PAINT workflow can be found in Appendix A. Each image-acquisition step was followed with three brief washing steps (1 min) of washing buffer and one washing step of imaging buffer (1 min). Then, the next imager strand solution (imager + imaging buffer) was introduced. The acquisition and washing steps were repeated until all the four targets were imaged.

#### 4.5.5. Drift-Correction Marker

Nitrogen-vacancy-center fluorescent nanodiamonds conjugated to streptavidin (FNDs; 40 nm; Adamas Nanotechnologies, Raleigh, NC, USA) were employed as reference markers for SMLM. Nanodiamonds stored in PBS with 0.1% BSA at 1 mg/mL (1% *w*/*v*) were diluted 1:50 and incubated with stained cells for 20’ at room temperature.

### 4.6. DNA PAINT Image Reconstruction

Single-molecule localization fitting was performed, with the Offline N-STORM Analysis module (NIS Elements software, version 5.42.01) correcting for spatial drift and chromatic aberrations. Before exporting DNA PAINT images, the Gaussian size of the single localizations and the format of the reconstructed images were set to optimize correlative imaging. We finally generated a dual-color DNA PAINT image of 41 × 41 μm (5 nm/px) with a Gaussian size of 10 nm. Localization precision was estimated in the range between 5 and 15 nm.

### 4.7. Automated Acquisition Protocol

The acquisition protocol was described in previous works [11]. Briefly, a low-resolution map of the cells was obtained by a 4× objective, and regions of interest were chosen according to the optimal density and minimal presence of large cell aggregates. The selected ROIs were then acquired, and data were analyzed with the A.M.I.CO analysis package described below to select the phenotype of interest. The software recalculated stage coordinates, allowing repositioning for SMLM analysis.

### 4.8. A.M.I.CO Image Analysis

The A.M.I.CO analysis package was developed within the open-source ImageJ (version 1.54b) platform using its macro programming language, as described in detail in a previous study [37]. The software is freely available upon request, or it can be downloaded from the GitHub public repository (https://github.com/MarioFaretta/AMICO, accessed on 20 November 2022). Briefly, the software performs cell identification by automated segmentation providing integrated and mean values per pixel of all the acquired fluorescence channels. The measured data were then reorganized by an analysis module to perform statistical calculation to identify cell subpopulations.

### 4.9. Correlative Microscopy

Initially, a widefield image, centered on the cell of interest, was acquired in the channel employed for both DNA PAINT and confocal imaging. Spatial sampling was set to satisfy the Nyquist criterion (65 nm pixel size, no binning). The digital size of the collected image was adjusted to obtain a similar field of view (FOV) between WF and single-molecule images, but with different spatial sampling (65 nm for WF, 160 nm for DNA PAINT). DNA PAINT images were in fact acquired by inserting a 0.4x Relay lens to enlarge the FOV. A digital-size 256 × 256 pixel image in a DNA PAINT image matched with 620 × 620 pixels of a WF ROI (65 nm/pixel). WF images were also used to check the potential drift in the pictures collected by the single-molecule acquisition and the confocal image.

For confocal imaging, the pixel size was adjusted to match the spatial sampling of the WF image (about 65 nm). The pinhole size was set to 0.6 Airy unit. Bidirectional galvanometric scanning was enabled and the confocal acquisition parameters, such as line average, pixel dwell time, laser power, detector filters, and gain, were adjusted to increase quality signal without saturation while also minimizing background signal and photobleaching.

Then, the 2-color DNA PAINT acquisition was run. Single-molecule analysis on the two acquired datasets identified the positions of the fluorescent reporter molecules to reconstruct the two DNA PAINT images, including a correction of stage drift during the acquisition. Final alignment of the WF, confocal, and DNA PAINT images was performed after acquisition. The confocal and high-resolution WF images were aligned using registration software (the plugin included in NIS software version 5.42.01 or the open-source TurboReg plugin in the ImageJ platform were employed in this work). The same process was repeated to align the drifted widefield image, acquired with the 0.4x Relay lens of the single-molecule acquisition module. The registration parameters were finally applied to obtain the aligned reconstructed DNA PAINT images ([11], Supplementary Protocol).

## Figures and Tables

**Figure 1 ijms-25-04672-f001:**
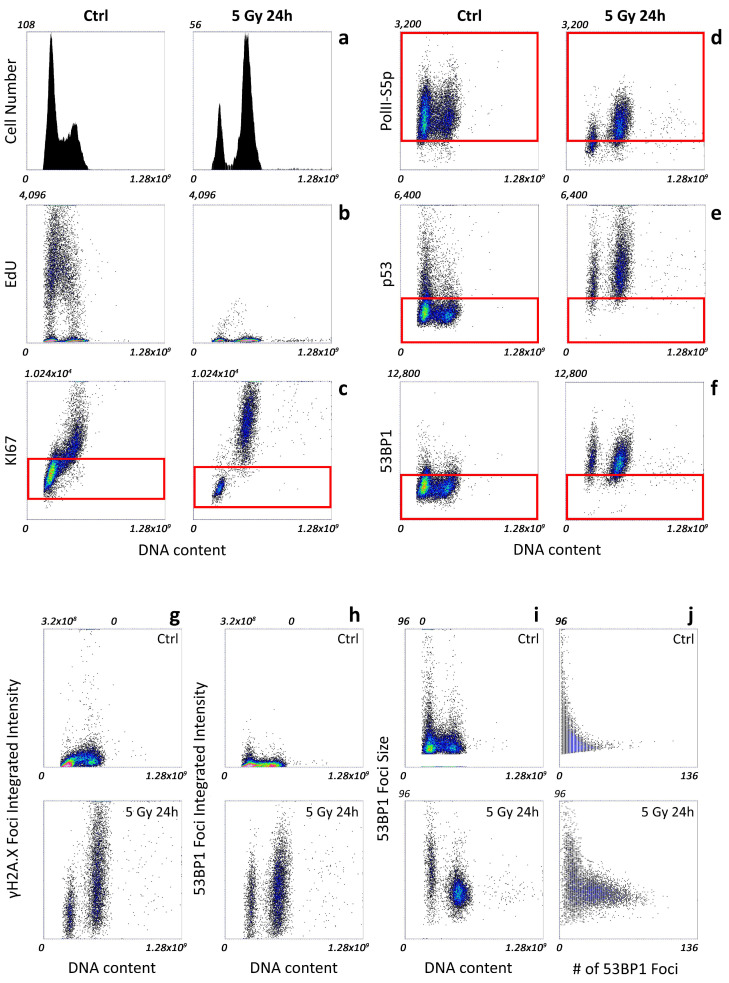
Image-cytometry analysis of a representative experiment of exponentially growing (n = 11399 cells) and X-ray irradiated (n = 6109) MCF10A cells showing the analysis of cell-cycle progression, proliferation state, transcriptional activity, p53 content measurement, and DDR response. (**a**) Histogram of DNA content distribution. (**b**) Dot plots of DNA content (X axis) versus EdU incorporation (Y axis) to mark active DNA replication. (**c**) Dot plots of DNA content (X axis) versus integrated intensity per cell nucleus of KI67 proliferation marker (Y axis). (**d**) Dot plots of DNA content (X axis) versus PolII-S5p mean intensity per pixel (Y axis). The red box was a guide drawn on the lower limit of the control-cell distribution approximated with Gaussian approximation. (**e**) Dot plots of DNA content (X axis) versus p53 mean intensity per pixel (Y axis). The red box was a guide drawn on the upper limit of the control-cell distribution approximated as a Gaussian approximation. (**f**) Dot plots of DNA content (X axis) versus nucleoplasm 53BP1 mean intensity per pixel (Y axis) calculated after subtraction of the mask corresponding to the identified DDR foci. The red box was a guide drawn on the upper limit of the control-cell distribution approximated as a Gaussian approximation. (**g**) Dot plots of DNA content (X axis) versus integrated intensity of the segmented γH2A.X foci per nucleus (Y axis). (**h**) Dot plots of DNA content (X axis) versus integrated intensity of the segmented γH2A.X foci per nucleus (Y axis). (**i**) Dot plots of DNA content (X axis) versus the average size (pixels) of the 53BP1 foci per nucleus (Y axis). (**j**) Dot plots of the number of 53BP1 foci (X axis) versus the average size (pixels) of the 53BP1 foci per nucleus (Y axis). Colors are representative of the density of the events.

**Figure 2 ijms-25-04672-f002:**
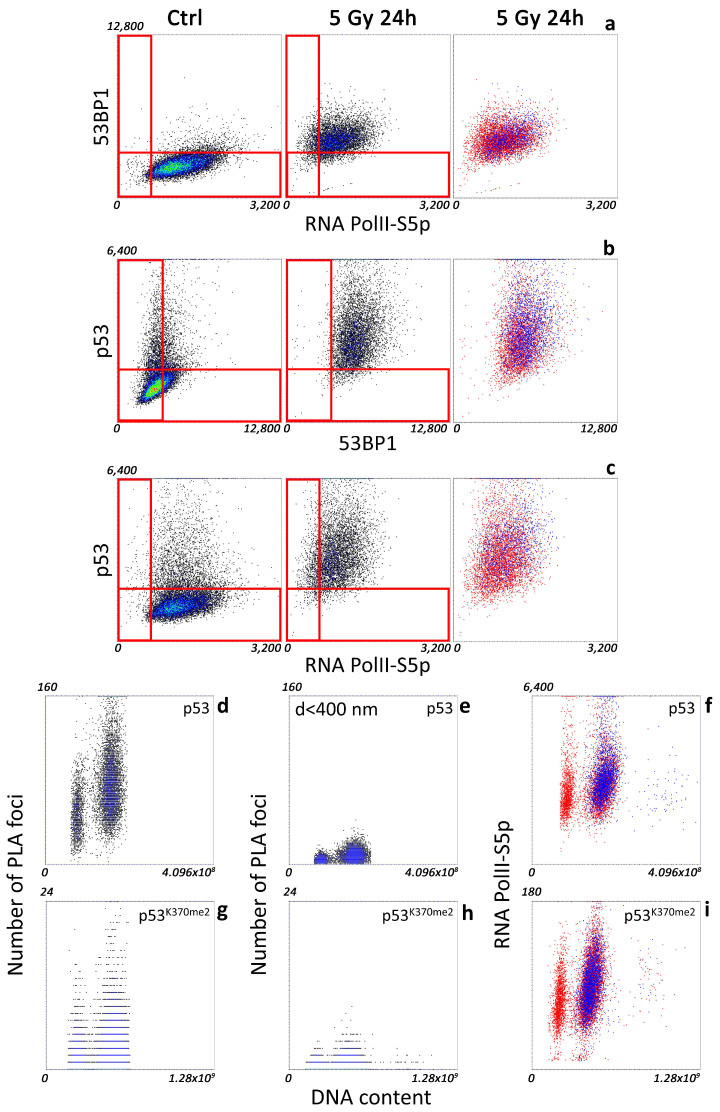
Image-cytometry analysis of a representative experiment of exponentially growing and X-ray-irradiated MCF10A cells. (**a**) RNAPolII-S5p versus 53BP1, (**b**) 53BP1 versus p53, and (**c**) RNAPolII-S5p versus p53 mean intensity per nucleus (n = 7100). The red box was a guide drawn on the lower (PolII-S5p) or upper (p53, 53BP1) limit of the control-cell distribution approximated with a Gaussian approximation. The third column shows the distribution of cells with the highest expression of p53 (**a**), PolII-S5p (**b**), and 53BP1 (**c**) (blue dots) over the entire population (red dots). (**d**) Dot plots of DNA content (X axis) versus the number of segmented p53-53BP1 PLA spots per nucleus (Y axis) 24 h after irradiation. (**e**) Dot plots of DNA content (X axis) versus the number of the segmented p53-53BP1 PLA spots per nucleus (Y axis) having at least one γH2A.X focus within 400 nm at 24 h after irradiation. (**f**) Dot plots of DNA content (X axis) versus PolII-S5p mean intensity per pixel (Y axis) showing the distribution of cells with the highest number of PLA spots per nucleus (blue dots) over the entire population (red dots). (**g**–**i**) Dot plots reporting the same info shown in (**d**–**f**) referred to a PLA analysis of p53 di-methylated on lysine 370 and 53BP1 (n = 10,228). Colors, except for the last column of Dot Plots, are representative of the density of the events.

**Figure 3 ijms-25-04672-f003:**
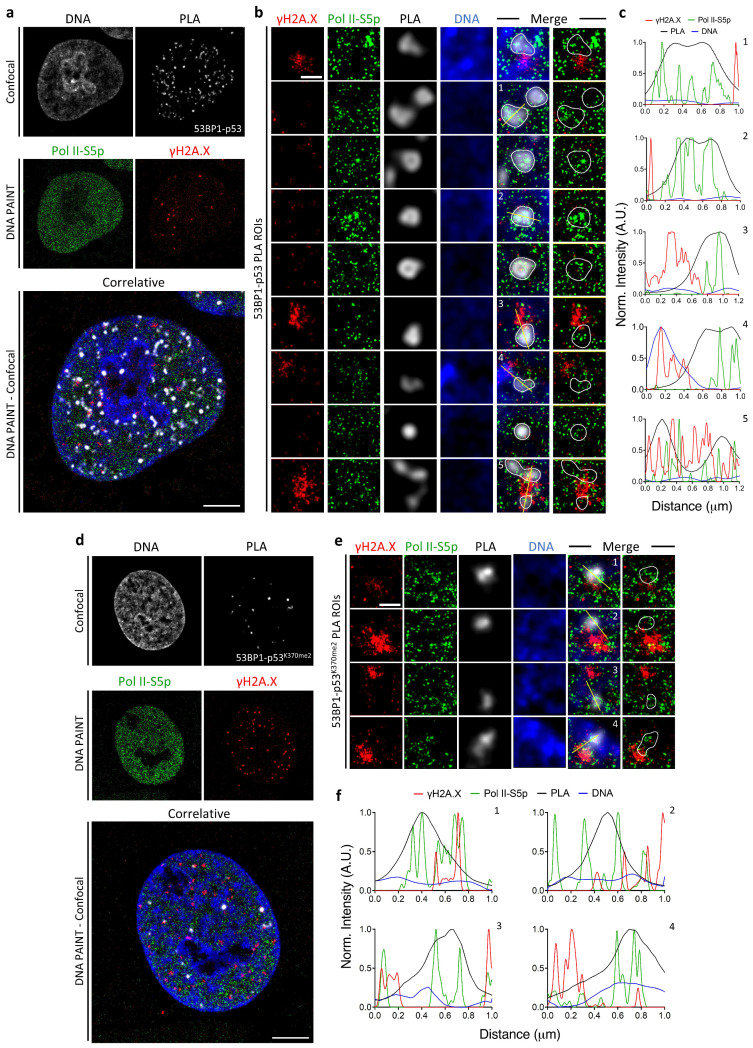
Correlative confocal-DNA PAINT imaging of X-ray-irradiated MCF10A nucleus. (**a**,**d**) Confocal images of Hoechst33342 (DNA) and PLA foci (53BP1-p53 (**a**) and 53BP1-p53^K370Me2^ (**d**)) were first acquired, followed by single-molecule acquisition of Polll-S5p (green) and yH2A.X (red) in DNA PAINT. Pol ll-S5p and γH2A.X were acquired using Cy3b and ATTO655 imagers, respectively. Two-color single-molecule image was then aligned and merged with confocal data. DNA PAINT acquisition parameters: 20,000 frames/ch, 90 ms exposure, 1 nM total imager concentration. Scale bar: 5 μm. (**b**,**e**) Representative ROIs showed the molecular distribution of Pol II-S5p (green) and γH2A.X (red) in single PLA spots of MCF10A cells exposed to irradiation (IR; 5 Gy). Scale bar: 500 nm. (**c**,**f**) Line profile analysis of the four signals (along the yellow line in the insets) showed colocalization of Pol II-S5p with PLA spots, in opposition to γH2A.X and Hoechst signals which did not show significative overlap.

**Figure 4 ijms-25-04672-f004:**
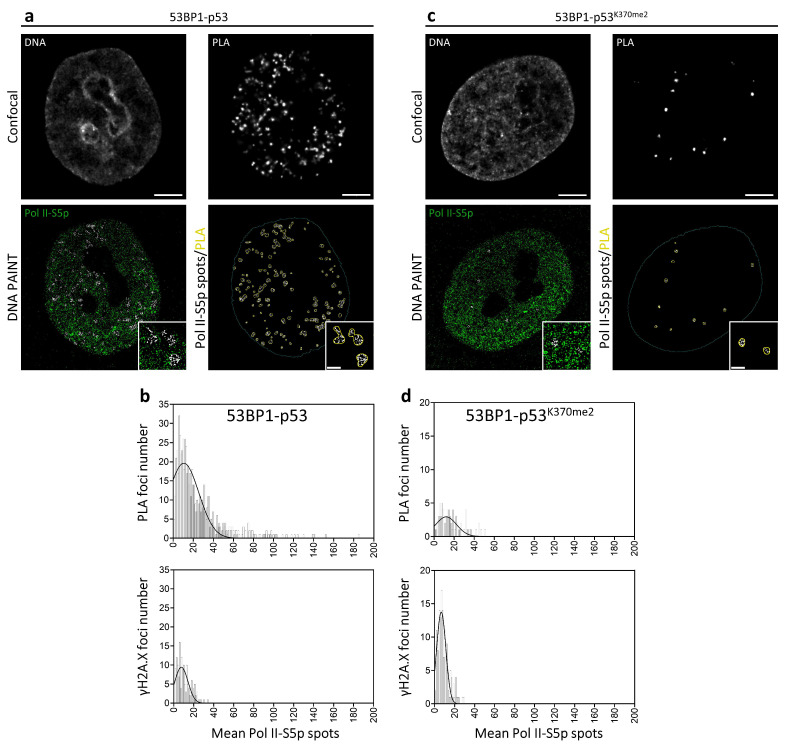
A.M.I.CO image analysis of Pol II-S5p single molecule events distribution at PLA on X-ray-irradiated MCF10A cells. Shown are (up to bottom) the confocal images of Hoechst33342 and PLA foci (53BP1-p53 (**a**) and 53BP1-p53^K370Me2^ (**c**)), DNA PAINT image of Pol II-S5p, and the segmentation of Pol II-S5p spots inside PLA foci. White spots represented the colocalized fraction of Pol II-S5p from the total number of detected molecules (green). (**b**,**d**) Total distribution of the number of PLA and γH2A.X foci containing the number of Pol II-S5p spots in both samples. Scale bar: 5 μm. Scale bar ROI: 1 μm.

**Figure 5 ijms-25-04672-f005:**
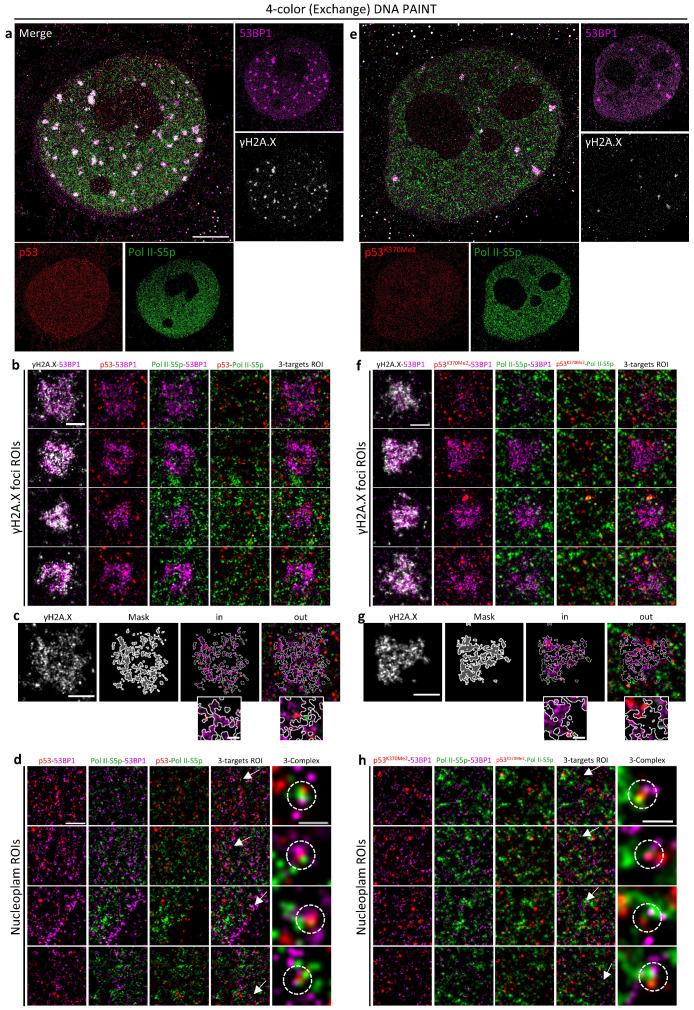
Analysis of p53, p53^K370Me2^, 53BP1, Pol II-S5p, and γH2A.X interaction at molecular level by 4-color Exchange PAINT of X-ray-irradiated MCF10A nuclei. (**a**,**e**) Shown is the single-molecule acquisition of 53BP1 (magenta), p53 (**a**) and p53^K370Me2^ (**e**) (red), Pol ll-S5p (green), and γH2A.X (white) of a representative cell. Target proteins were sequentially acquired using Cy3b and ATT0655 imagers. Imaging parameters: 20,000 frames/ch, 90 ms exposure, 1 nM total imager concentration. Scale bar: 5 μm. (**b**,**f**) Representative ROIs showed the molecular distribution of 53BP1 (magenta), p53 (**b**) and p53^K370Me2^ (**f**) (red), and Pol ll-S5p (green) in IR-induced DNA damage foci detected by γH2A.X (white) (IR; 5 Gy). The segmentation analysis of a representative γH2A.X focus (**c**,**g**) showed relevant colocalization with 53BP1 signal (in); nevertheless, 53BP1 and γH2A.X foci evinced different spatial distributions (**b**,**f**). p53, p53^K370Me2^, and Pol ll-S5p did not show a significative overlap (out), in agreement with the downregulation of the transcriptional program during the DNA damage and repair processes. Scale bar: 500 nm, 100 nm. (**d**,**h**) Representative ROIs showed the molecular distribution of 53BP1, p53, p53^K370Me2^, and Pol ll-S5p in nucleoplasm. In both the p53 and p53^K370Me2^ samples, multi-color DNA PAINT revealed a proximity between protein targets (white arrows), suggesting a putative three-complex between Pol ll-S5p, 53BP1 and two p53 isoforms (white dot circles). Scale bar: 500 nm, 100 nm.

**Figure 6 ijms-25-04672-f006:**
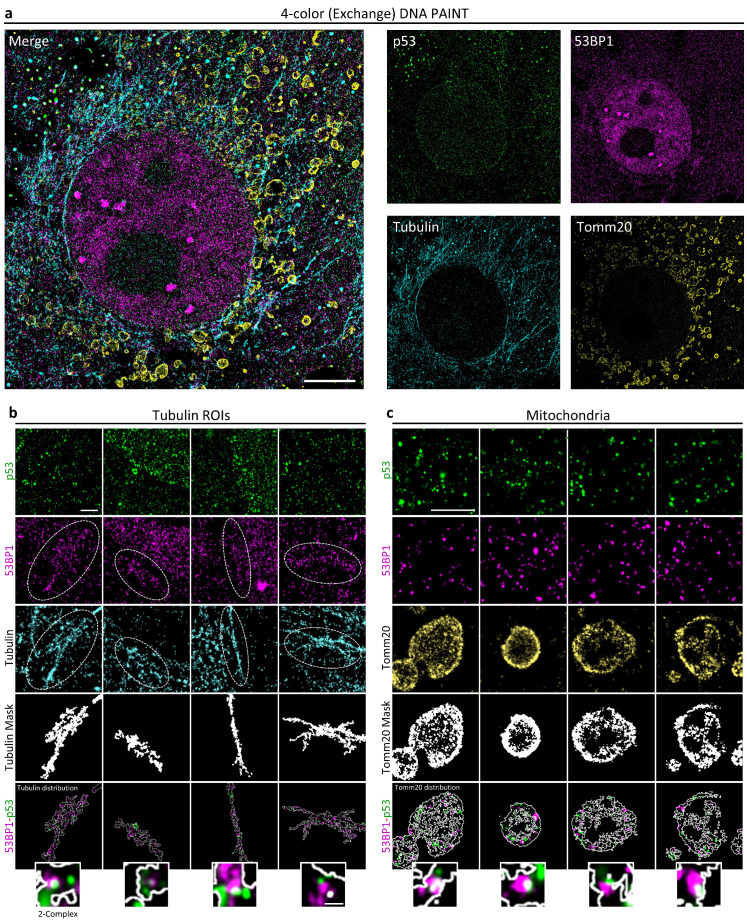
Analysis of molecular distribution of 53BP1 and p53 in cytoskeletal (β-Tubulin) and mitochondrial (Tomm20) compartments on X-ray-irradiated MCF10A cells. (**a**) Shown is the 4-color Exchange PAINT image of 53BP1 (magenta), p53 (green), Tomm20 (yellow), and β-Tubulin (cyan) of a representative cell. Scale bar: 5 μm. (**b**) Tubulin ROIs showed the molecular distribution of p53 (green) and 53BP1 (magenta) on β-Tubulin signal (cyan). Shown is a significant accumulation of 53BP1 signal on a β-Tubulin structure (white dot ellipses). The segmentation analysis of β-Tubulin filaments revealed a relevant colocalization with a 53BP1 signal. Interestingly, multi-color DNA PAINT detected 53BP1-p53 interaction alongside the β-Tubulin signal. (**c**) Mitochondria ROIs showed the molecular distribution of p53 (green) and 53BP1 (magenta) on the Tomm20 signal (yellow). As a cytoskeletal compartment, the segmentation analysis of the Tomm20 signal showed off a colocalization between 53BP1-p53 in mitochondria. Scale bar: 1 μm, 100 nm.

## Data Availability

Raw data are available upon request.

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
