# Peer review of "From Cell Populations to Molecular Complexes: Multiplexed Multimodal Microscopy to Explore p53-53BP1 Molecular Interaction"

_ijms, 2024, doi:10.3390/ijms25094672_

Round 1

Reviewer 1 Report

Comments and Suggestions for Authors
  1. Specify the optical performance of the proposed technologies: spatial resolution, depth resolution, FOV, SNR, imaging speed; Specify the system parameters: Irradiance at the sample plane; integration time for each pixel and each image. Especially, what is the photon budget considering single molecular imaging

  1. When imaging fluorescence, did the author deal with photobleaching and phototoxicity? 

  1. In Figure 6, what algorithm is used in the image segmentation? Can the author discuss the method in detail? 

  1. The authors mentioned spatio-temporal dynamics. How to further increase imaging speed to resolve fast dynamics?

  1. Fluorescence lifetime imaging microscopy(FLIM) is an important molecular imaging method. Compared with intensity-based methods, lifetime-based methods offer more robustness. References below are several FILM-based methodologies which could serve as valuable references in this context. 

Can the author discuss the potential of using FLIM to track different molecules? 

[1] https://doi.org/10.1007/s10585-008-9204-0

[2] https://doi.org/10.1073/pnas.2004176118

[3] https://doi.org/10.1364/OPTICA.5.001290

Author Response

  1. Specify the optical performance of the proposed technologies: spatial resolution, depth resolution, FOV, SNR, imaging speed; Specify the system parameters: Irradiance at the sample plane; integration time for each pixel and each image. Especially, what is the photon budget considering single molecular imaging.

We thank the reviewer for noting the absence of this data. We amended the text in the Material and Methods by inserting the requested info.

  1. When imaging fluorescence, did the author deal with photobleaching and phototoxicity?

One of the advantages offered by automated multimodal microscopy is the possibility to adopt the most convenient imaging conditions including the minimization of photobleaching (phototoxicity is not an issue in this work being focused on fixed samples. However, the same considerations can be extended to the in vivo situation). The initial widefield observation for the target identification is extremely fast and operated under conservative conditions (optimal exposure-time versus illumination-power). Confocal acquisition is generally performed by resonance scanning. Finally, photobleaching minimization during DNA-PAINT analysis is obtained by an oxygen scavenging system in the buffer (Schnitzbauer, J., et al. (2017). "Super-resolution microscopy with DNA-PAINT." Nat Protoc 12(6): 1198-1228) and excitation power adjustment. More than this, in DNA-PAINT the use of an inclined illumination and the large excess of fluorophore granted by the reservoir of reporter molecules in the buffer (we also employed large volumes to minimize evaporation during the long time-lapse imaging phase) preserved a constant ratio of recognized events during the experiment.

  1. In Figure 6, what algorithm is used in the image segmentation? Can the author discuss the method in detail?

Segmentation of the region in figure 6 has been applied to the clustered reconstruction of the localized events in the indicated channels. Regions have been selected by the ImageJ magic wand selection tool with 8-connected tolerance applying a threshold calculated by the Li algorithm present in the software.

  1. The authors mentioned spatio-temporal dynamics. How to further increase imaging speed to resolve fast dynamics?

We thank the reviewer for the observation. In the presented work we focused on an example of frozen interactions by working on fixed material. The temporal view is contained in the large statistical sampling granted by the first widefield acquisition of the cell population, as discussed in the text, thanks to the identification of different phenotypes observed in the DNA Damage Response. Live cell imaging can also be inserted in the context of automated acquisition-driven microscopy to enable fast molecular dynamics and we inserted in the discussion some references on how to optimize temporal resolution to enable its analysis (see the point below).

  1. Fluorescence lifetime imaging microscopy (FLIM) is an important molecular imaging method. Compared with intensity-based methods, lifetime-based methods offer more robustness. References below are several FILM-based methodologies which could serve as valuable references in this context. Can the author discuss the potential of using FLIM to track different molecules?

We thank the reviewer for the useful suggestion: we expanded the Discussion (lines 529-554) by mentioning the adaptation to in vivo situations of the analysis-driven acquisition also reporting the great potential of FLIM in its time-compressed derivatives (we thank the reviewer for the useful references, since our group is not familiar with this technology), particularly for label-free analysis of fluorescent endogenous molecules.

Reviewer 2 Report

Comments and Suggestions for Authors

The manuscript presents a single molecule localization microscopy study with a resolution of tens of nanometers. The authors clearly stated the obstacles in the field of microscopy for imaging biological samples with optimal spatial resolution. Using an automated analysis-driven acquisition with an adapted Exchange-PAINT protocol, the authors visualized the formation of a putative macromolecular complex and demonstrated the localization of a single molecule with high precision. They specifically designed a pipeline that used proximity ligation analysis to visualize in the range of 50 nm and combined it with diffraction-limited confocal microscopy and single molecule localization microscopy. The effectiveness of the pipeline has been proved by the study of the putative interaction between p53 and 53PB1 protein, where the localization has a precision smaller than 20 nm.  I have the following questions and comments about the manuscript:

1.      The manuscript started by mentioning different variants of single-molecule localization microscopes. It would be beneficial if the authors give some brief introductions about different techniques in terms of advantages and disadvantages.

2.      The manuscript highlights the potential of automated multi-modal multi-resolution microscopy to transform biomedical research. How might these techniques be applied to ongoing challenges in cellular biology, such as tracking transient molecular interactions or studying the dynamics of cellular processes in real-time?

3.      The labels in the figures, especially for Figure 3, are generally too small to be seen.

Author Response

  1. The manuscript started by mentioning different variants of single-molecule localization microscopes. It would be beneficial if the authors give some brief introductions about different techniques in terms of advantages and disadvantages.

We thank the reviewer for the useful suggestion: we modified the introduction (lines 38-81) by inserting a brief description of the available single molecule techniques.

  1. The manuscript highlights the potential of automated multi-modal multi-resolution microscopy to transform biomedical research. How might these techniques be applied to ongoing challenges in cellular biology, such as tracking transient molecular interactions or studying the dynamics of cellular processes in real-time?

According to the reviewer request we introduced in the discussion a comment (lines 529-554) on the potential of automated multimodal microscopy including references to the possible applications in the study of in vivo molecular dynamics and interactions.

  1. The labels in the figures, especially for Figure 3, are generally too small to be seen.

 Thanks for the observation: we modified the figures with enlarged labels finding a compromise between figure compactness and readability due to the large number of subpanels. We checked that with page width zoom all the labels maintained their visibility.